# APDDv2: Aesthetics of Paintings and Drawings Dataset with Artist Labeled Scores and Comments

**Xin Jin[1,2], Qianqian Qiao[1], Yi Lu[3], Huaye Wang[1], Heng Huang[*4], Shan Gao[5], Jianfei Liu[3], and Rui Li[3]**

[1]**Beijing Electronic Science and Technology Institute**
[2]**Beijing Institute for General Artificial Intelligence**
[3]**Central Academy of Fine Arts**
[4]**University of Science and Technology of China**
[5]**Beijing University of Technology**

## Abstract

Datasets play a pivotal role in training visual models, facilitating the development of abstract understandings of visual features through diverse image samples and multidimensional attributes. However, in the realm of aesthetic evaluation of artistic images, datasets remain relatively scarce. Existing painting datasets are often characterized by limited scoring dimensions and insufficient annotations, thereby constraining the advancement and application of automatic aesthetic evaluation methods in the domain of painting. To bridge this gap, we introduce the Aesthetics Paintings and Drawings Dataset (APDD), the first comprehensive collection of paintings encompassing 24 distinct artistic categories and 10 aesthetic attributes. Building upon the initial release of APDDv1[Jin et al., 2024], our ongoing research has identified opportunities for enhancement in data scale and annotation precision. Consequently, APDDv2 boasts an expanded image corpus and improved annotation quality, featuring detailed language comments to better cater to the needs of both researchers and practitioners seeking high-quality painting datasets. Furthermore, we present an updated version of the Art Assessment Network for Specific Painting Styles, denoted as ArtCLIP. Experimental validation demonstrates the superior performance of this revised model in the realm of aesthetic evaluation, surpassing its predecessor in accuracy and efficacy. The dataset and model are available at https://github.com/BestiVictory/APDDv2.git.

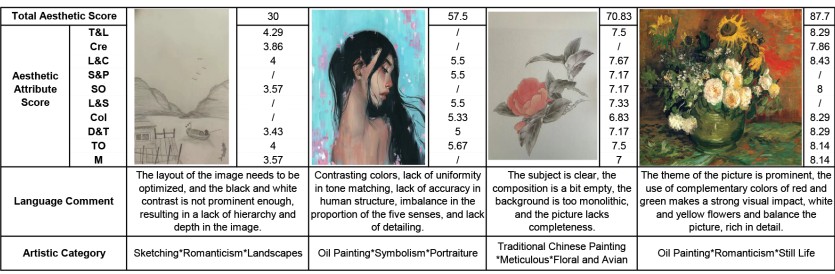

Figure 1: Samples from the APDDv2 dataset.

---

*Corresponding author: Heng Huang (hecate@mail.ustc.edu.cn)

# 1 Introduction

The IAQA (Image Aesthetic Quality Assessment) task aims to automatically evaluate the aesthetic quality of images through computer vision techniques. IAQA can be summarized into five layers of tasks as illustrated in the Figure 2 [Jin et al., 2018]. In the IAQA task, the preponderance of datasets is concentrated within the realm of photography. If the APDD dataset is not considered, across the initial three phases of the IAQA task, the quantity of aesthetic image datasets within the photography domain totals a minimum of 685,000 images, contrasting sharply with the scanty 60,000 images found in the painting domain. Within the Aesthetic Attributes phase, datasets sourced from the photography domain comprise no fewer than 31,000 images, while their painting counterparts consist of a mere 4,248 images. Moving to the Aesthetic Captions phase, datasets originating from the photography domain encompass no less than 198,000 images, whereas there are no publicly available datasets specifically tailored for the painting domain.

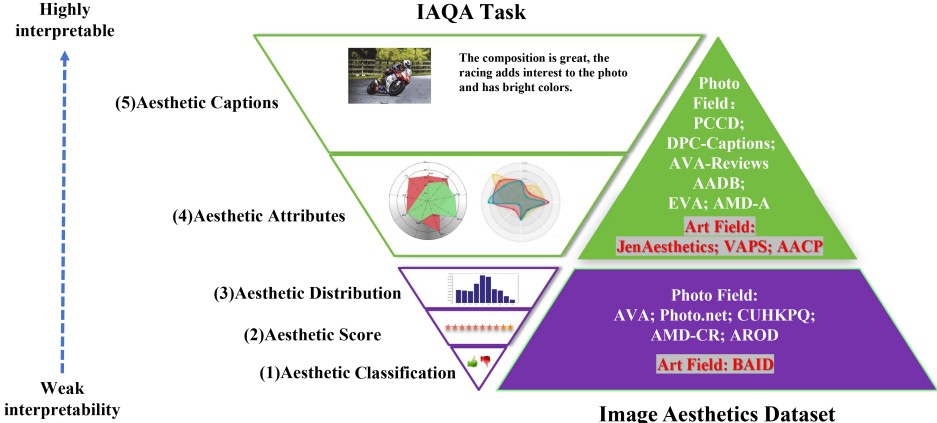

Figure 2: The five-layered tasks of IAQA exhibit an overall inverted triangular distribution in terms of corresponding data volume: as the hierarchy ascends, the data volume decreases, accompanied by a decrease in annotation quality.

Constructing high-quality benchmark datasets for paintings is crucial for advancing computer vision aesthetics in art, providing a framework for analyzing and understanding artistic imagery. Unlike the uniform realm of photography, the diversity and intricacies of artistic expression significantly heighten the challenge of dataset construction, requiring meticulous consideration of variations in styles, themes, and techniques. The subjective nature of aesthetic evaluations in art necessitates accounting for the artist's intent and audience perception, demanding professional expertise. In collaboration with experts and educators, we established a robust system for assessing aesthetic components, categorizing paintings into 24 art categories and 10 aesthetic attributes, with defined scoring standards. This effort resulted in APDDv1[Jin et al., 2024], comprising 4,985 images and over 31,100 annotations from 28 art experts and 24 art students, with each image evaluated by at least six annotators, covering both overall aesthetic scores and specific attribute scores.

To further enhance the dataset's quality and richness, we assembled a more specialized labeling team and introduced aesthetic language comments, leading to the release of APDDv2. This updated version expands the image count to 10,023 and the number of annotations exceeds 90,000, including detailed aesthetic comments. Building on the foundation of APDDv1, we developed more detailed and applicable scoring standards for artistic images. These enhancements aim to create a more comprehensive, rich, and high-quality dataset for the aesthetics of paintings and sketches, thereby providing a more reliable foundation for the analysis and research of artistic images.

The primary goal of constructing our dataset is to train aesthetic models. To enhance the CLIP image encoder [Radford et al., 2021] for image aesthetics, we employed a fine-tuning approach tailored for this domain. Directly applying the original CLIP to Image Aesthetic Analysis (IAA)

yields suboptimal results due to the generic nature of datasets like COCO[Lin et al., 2014] and Flickr30K[Plummer et al., 2015], which lack the distinct features influencing aesthetic perception. To address this, AesCLIP [Sheng et al., 2023] uses contrastive learning to capture representations sensitive to aesthetic attributes, bridging the gap between general and aesthetic image domains. Drawing inspiration from the AesCLIP approach, we trained ArtCLIP on the APDDv2 dataset. Experimental results show that ArtCLIP surpasses state-of-the-art techniques, enhancing both image aesthetic analysis and related research tools.

Overall, our contributions include:

We developed a comprehensive set of evaluation criteria specifically tailored for artistic images, effectively assessing their aesthetic components. Furthermore, we provided a practical and scalable evaluation methodology for the continuous expansion of the dataset.

We introduced the APDDv2 dataset (the license of the dataset: CC BY 4.0), created with the participation of over 40 experts from the painting domain, comprising 10,023 images, 85,191 scoring annotations, and 6,249 language comments.

We developed ArtCLIP, an art assessment network for specific painting styles, utilizing a multi-attribute contrastive learning framework. Experimental validation demonstrated that the updated model outperforms existing techniques in aesthetic evaluation.

## 2 Related Work

### 2.1 Artistic Image Aesthetic Assessment Datasets

Table 1: A comparison between the APDD dataset and existing image datasets.

| Dataset | Number of Images | Number of Attributes | Number of Categories | Any Comment? |
|---------|------------------|----------------------|----------------------|--------------|
| BAID  [Yi et al., 2023] | 60,337 | - | - | NO |
| AACP  [Jiang et al., 2024] | 21,200 | - | - | NO |
| VAPS  [Fekete et al., 2022] | 999 | 5 | 5 | NO |
| JenAesthetics  [Amirshahi et al., 2015] | 1,268 | 5 | 16 | NO |
| JenAesthetics$\beta$  [Amirshahi et al., 2016] | 281 | 1 (beauty) | 16 | NO |
| MART  [Yanulevskaya et al., 2012] | 500 | 1 (emotion) | - | NO |
| APDDv1  [Jin et al., 2024] | 4,985 | 10 | 24 | NO |
| **APDDv2** | **10,023** | **10** | **24** | **YES** |

In the realm of artistic image aesthetics, some datasets have been introduced. The BAID dataset[Yi et al., 2023], unveiled in 2023, comprises 60,337 high-quality art images sourced from the Boldbrush website[2], with each image accompanied by an aesthetic score derived from user votes. Released in 2024, the AACP dataset[Jiang et al., 2024] encompasses 21,200 children's drawings, among which 20,000 unlabeled images are generated by the DALL-E model based on keyword combinations, while 1,200 images are scored for eight attributes by 10 experts. The VAPS dataset[Fekete et al., 2022], launched in 2022, showcases 999 representative works from the history of painting art, categorized into five image classes, with attribute scores assigned from five perspectives for each image. The JenAesthetics$\beta$ dataset[Amirshahi et al., 2016], unveiled in 2013, showcases 281 high-quality color paintings sourced from 36 diverse artists. The JenAesthetics dataset[Amirshahi et al., 2015], released in 2015, boasts 1,628 color oil paintings exhibited in museums. Lastly, the MART dataset[Yanulevskaya et al., 2012], introduced in 2012, curates 500 abstract paintings from art museums, each rated with positive or negative emotion.

### 2.2 Artistic Image Aesthetic Assessment Models

Benefiting from the establishment and utilization of large-scale datasets, the field of Image Aesthetic Analysis (IAA) has undergone a significant evolution. It has transitioned from its early reliance on manually designed features and traditional machine learning methods to the current era driven by deep learning technologies, which has led to the emergence of numerous IAA models. However,

---

[2]https://artists.boldbrush.com/

within the domain of painting, there remains a scarcity of IAA models. The SAAN model[Yi et al., 2023], introduced in 2023, aims to assess the aesthetic appeal of art images by integrating techniques such as self-supervised learning, adaptive instance normalization, and spatial information fusion. This integration allows for the effective utilization of artistic style and general aesthetic features, resulting in accurate assessments of aesthetic appeal in art images. On the other hand, the AACP model[Jiang et al., 2024], unveiled in 2024, employs a self-supervised learning strategy to address the issue of insufficient data annotation. It leverages spatial perception networks and channel perception networks to retain and extract spatial and channel information from children's drawings. The objective of AACP is to more precisely evaluate the aesthetic appeal of children's drawings, thereby fostering the development of children's art and aesthetic education. Our previously proposed AANSPS model[Jin et al., 2024] utilizes EfficientNet-B4 as the backbone network and incorporates efficient channel attention (ECA) modules and regression networks to evaluate the overall score and attribute scores of art images.

## 3 APDDv2

### 3.1 Artistic Categories and Aesthetic Attributes

As shown in Figure 3, the APDD dataset has been categorized into 24 distinct artistic categories[Jin et al., 2024] based on painting type, artistic style, and subject matter.

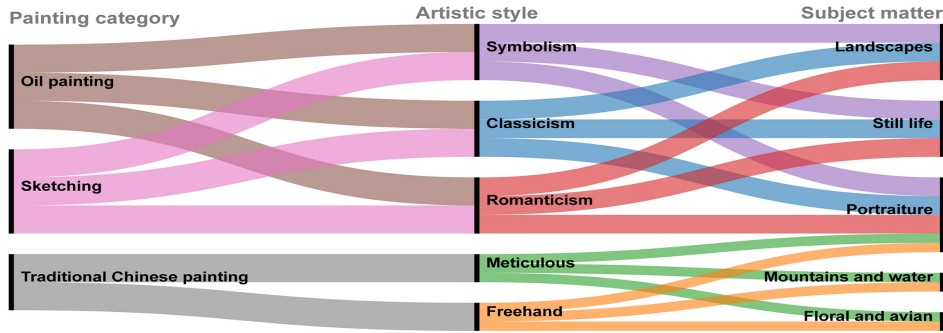

Figure 3: 24 Artistic Categories in the APDD Dataset.

Table 2: 10 Aesthetic Attributes of APDDv2

| Aesthetic Attribute | Abbreviation | Interpretation |
| --- | --- | --- |
| Theme and Logic | T&L | The central idea aligns with the artistic expression, ensuring consistency and appropriateness in composition, layout, and color. |
| Creativity | Cre | Innovative qualities that break conventions, including satire, self-deprecation, and allegorical warnings. |
| Layout and Composition | L&C | The visual structure and organization of an image, reflecting the underlying logic and essence of its form. |
| Space and Perspective | S&P | Layered spatial arrangements and perspective techniques create three-dimensionality and spatial effects. |
| Sense of Order | SO | Visual unity and consistency in morphological, spatial, orientational, and dynamic elements. |
| Light and Shadow | L&S | Enhance visual rhythm and realism, decorate space, suggest themes, and segment the image. |
| Color | Col | Evokes emotional atmospheres with a harmonious palette, using contrasts in temperature, brightness, and purity. |
| Details and Texture | D&T | Vivid details and delicate textures enhance realism, imbuing life into the image. |
| The Overall | TO | Emphasizes coherence and a clear theme, combining form and spirit in the presentation. |
| Mood | M | Creates a poetic space blending scenes, reality, and illusion, emphasizing tranquility, emptiness, and spirituality. |

Through extensive collaboration with professional artists from the Central Academy of Fine Arts, we identified 10 aesthetic attributes[Jin et al., 2024]. These attributes are derived from the cognitive

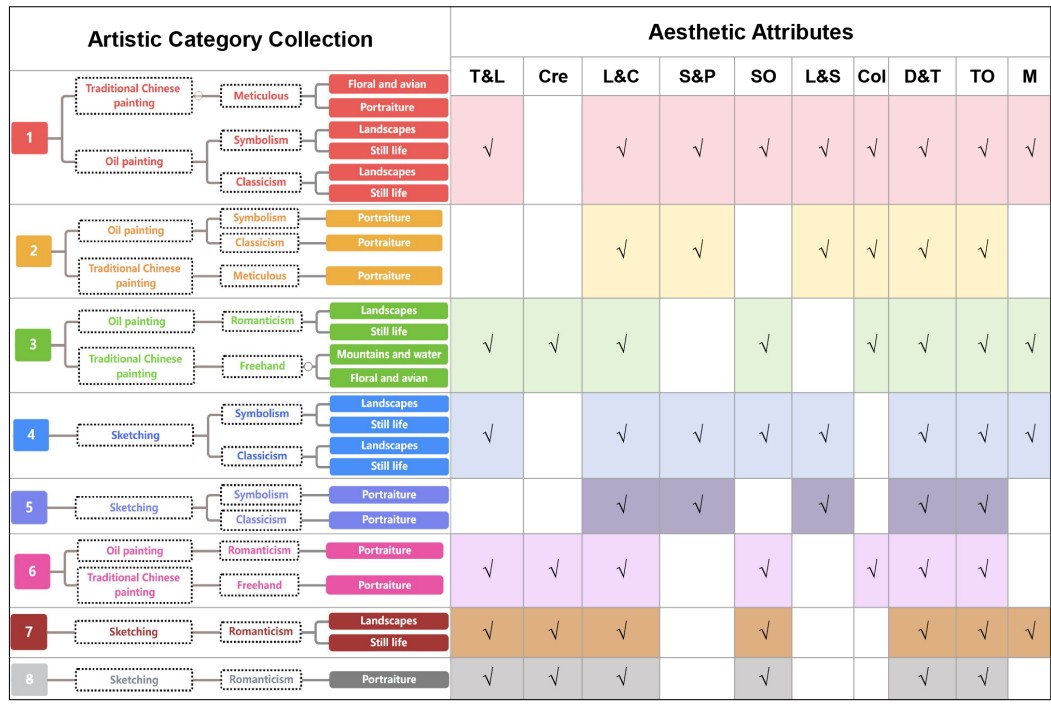

Figure 4: Correspondence between artistic categories and aesthetic attributes.

processes of artists, the layered observation methods of viewers, and established painting evaluation standards. The attributes are as Table 2.

Art is inherently diverse and subjective. The aesthetic attributes of artistic images are influenced by cultural backgrounds, historical periods, and the unique styles and philosophies of artists. Consequently, different types of paintings exhibit distinct aesthetic attributes. In the APDD dataset, the relationships between art types and these aesthetic attributes are illustrated in Figure 4.

To ensure a broad and diverse collection of artistic images, we meticulously curated data from various professional art websites and institutions. Key sources included ConceptArt[3], DeviantArt[4], WikiArt[Phillips and Mackintosh, 2011], and the China Art Museum[5]. The artworks obtained from these platforms typically demonstrated aesthetic qualities ranging from moderate to high. However, to further enrich the dataset's diversity and representativeness, and to maintain a balanced spectrum of aesthetic standards, we incorporated a portion of student works with varying aesthetic quality. These student contributions, constituting approximately one-fourth of the total dataset, were primarily sourced from the Affiliated Middle School of the Central Academy of Fine Arts, supplemented by submissions from mobile applications like "MeiYuanBang"[6] and "XiaoHongShu"[7].

Ultimately, we successfully expanded the dataset to include 10,023 paintings covering 24 categories, with each category containing at least 410 images. Through this approach, we effectively balanced high and low-rated artworks in the dataset, providing richer and more comprehensive materials for subsequent research and analysis.

---

[3] https://conceptartworld.com/
[4] https://www.deviantart.com/
[5] https://www.namoc.org/zgmsg/index.shtml
[6] https://annefigo.meiyuanbang.com/m/app
[7] https://www.xiaohongshu.com/explore

## 3.2 Labeling Team

With the aim of improving the quality and precision of annotations in APDDv2 and recognizing the necessity for additional linguistic insights, we opted to enlist more authoritative experts into the labeling process. Ultimately, the labeling team for APDDv2 comprised 37 individuals from diverse professional backgrounds. We organized this team into three groups based on their areas of expertise and personal preferences: the Oil Painting Group, the Sketching Group, and the Traditional Chinese Painting Group. Each group was led by the individual with the deepest qualifications and highest level of expertise within the group. As a result, we formed an labeling team consisting of 18 members in the Oil Painting Group, 10 members in the Sketching Group, and 9 members in the Traditional Chinese Painting Group.

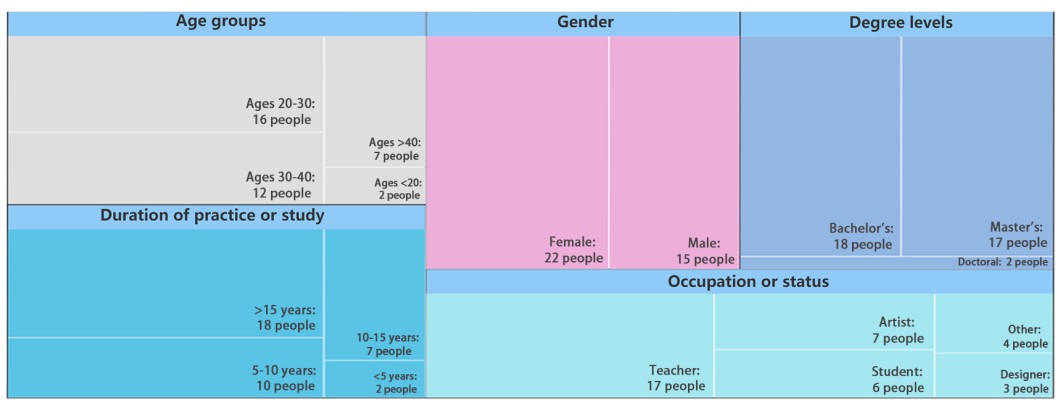

Figure 5: Labeling Team Composition.

As shown in Figure 5, the composition of the labeling team for APDDv2 encompasses a diverse array of educational backgrounds, professional identities, durations of practice or study, and age ranges. Notably, all annotators possess at least an undergraduate degree, ensuring a solid theoretical foundation and professional competence. Most team members have at least 10 years of work or study experience, which guarantees extensive practical expertise and profound professional knowledge in the painting domain. The age range of the annotators spans from teenagers to individuals in their 40s, providing the team with a broad spectrum of perspectives and diverse viewpoints. All annotators are focused on the field of painting, with half of them being professional art teachers who bring extensive teaching experience and art theory knowledge. The team also includes artists, designers, and students, ensuring diversity and comprehensiveness in the annotation work, thereby making the results more representative and authoritative.

## 3.3 Labeling Process

The APDD dataset comprises 24 categories of images, each featuring 5-9 attributes with corresponding scores for each attribute of every image. To establish a continuous evaluation system and maintain consistent standards between the two versions of the APDD dataset, we analyzed and summarized the annotation scores from APDDv1 and utilized these insights to define the scoring standards for APDDv2. Specifically, we selected images representing different score ranges for each attribute within each category, creating benchmark tables for all 24 categories. Figure 6 illustrates the benchmark table for the "Oil Painting - Symbolism - Still Life" category. In this table, we list the various attributes of the category images along with their score ranges. Subsequently, we selected representative images for each attribute score range to serve as benchmark images. These benchmark images exemplify the characteristics and qualities of different score ranges, assisting the labeling team in better understanding and applying the scoring standards.

This continuous evaluation system helps maintain the consistency and comparability of dataset annotations, allowing for coherent comparison and analysis across different versions of the dataset. It

enhances the quality and usability of the dataset by ensuring that annotations are made according to uniform standards.

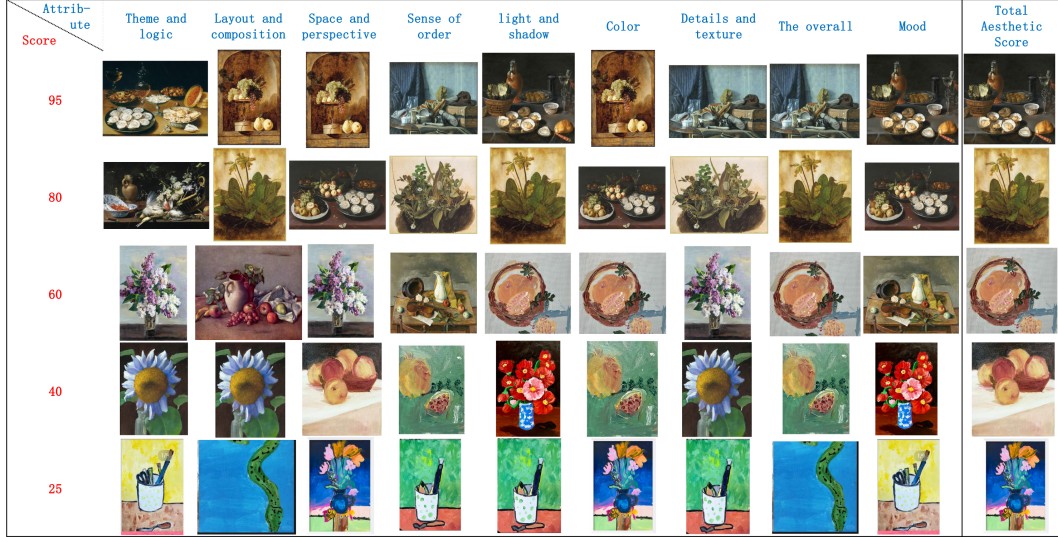

Figure 6: Scoring benchmark table for "Oil Painting - Symbolism - Still Life" category.

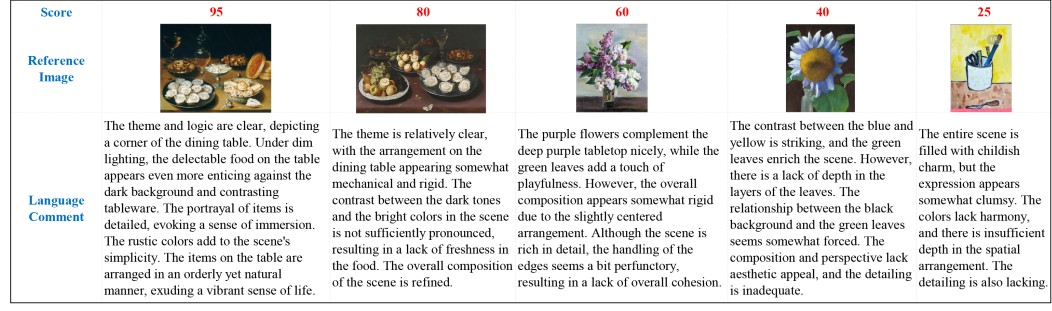

Figure 7: Benchmark table for language comments in "Oil Painting - Symbolism - Still Life" category.

In addition to the scoring benchmark table, APDDv2 necessitates the inclusion of language comments. Given that different types of art encompass distinct aesthetic standards, modes of expression, and artistic characteristics, it is essential to provide specific comment examples for each category of images. To achieve this, the group leaders, who are the most experienced and skilled members, provide comments that serve as crucial references. These examples not only assist annotators in comprehending the requirements of the annotation tasks but also offer concrete and actionable guidance. This helps ensure that the labeling team delivers consistent and accurate comments for different types of art. Consequently, we invited the team leaders to provide comment examples for each image type and score range. Figure 7 illustrates the comment standard example for the "Oil Painting - Symbolism - Still Life" category, as completed by the leader of the Oil Painting group.

The scoring benchmark image tables and comment examples for the 24 image categories provided the labeling team with clear reference standards and operational guidelines. To ensure a smooth and systematic labeling process, we assigned specific image categories and quantities to each team member, ensuring that each image was rated by at least six annotators and commented on by at least one annotator. The entire labeling process was conducted within a dedicated labeling system[8] specifically developed for APDD, and the task was completed within 15 days.

---

[8]http://103.30.78.19/

Table 3: Number of labels for each score type of APDDv2

| Score type | Total Aesthetic Score | Theme and Logic | Creativity | Layout and composition | Space and Perspective | |
|---|---|---|---|---|---|---|
| pre-averaging | 62,790 | 49,967 | 24,122 | 62,790 | 38,668 | |
| after averaging | 10,023 | 7,965 | 3,820 | 10,023 | 6,205 | |
| Score type | Sense of Order | Light and Shadow | Color | Details and Texture | The Overall | Mood |
| pre-averaging | 49,967 | 38,644 | 38,870 | 62,790 | 62,790 | 42,115 |
| after averaging | 7,965 | 6,205 | 6,202 | 10,023 | 10,023 | 6,737 |

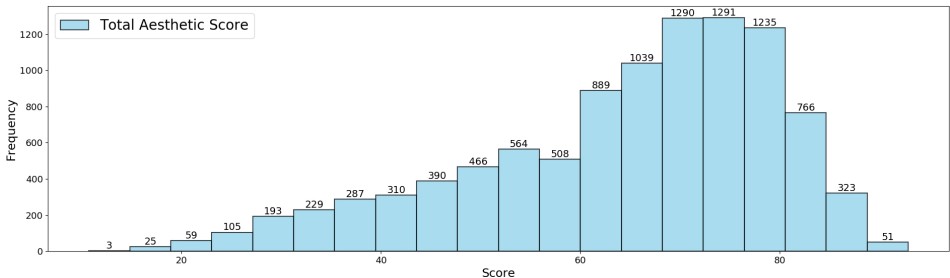

Figure 8: Distribution of Total Aesthetic Score. Most images received scores between 50 and 70, while images with extreme ratings were relatively rare, aligning with the general principles of human aesthetic perception.

After backend data cleaning, we amassed a total of 533,513 annotation records, encompassing 6249 comment records. It's noteworthy that the total aesthetic score ranges from 0 to 100, while the aesthetic attribute scores range from 0 to 10. However, the total aesthetic score isn't a mere sum of the aesthetic attribute scores because it typically accounts for the overall artistic effect and perception, rather than just aggregating individual attribute scores. Finally, through averaging the scores assigned to each image, we calculated the final score for each attribute, alongside the total aesthetic score for each image. Table 3 presents a comparison of the number of labels in the APDDv2 dataset before and after averaging the scores.

# 4 ArtCLIP

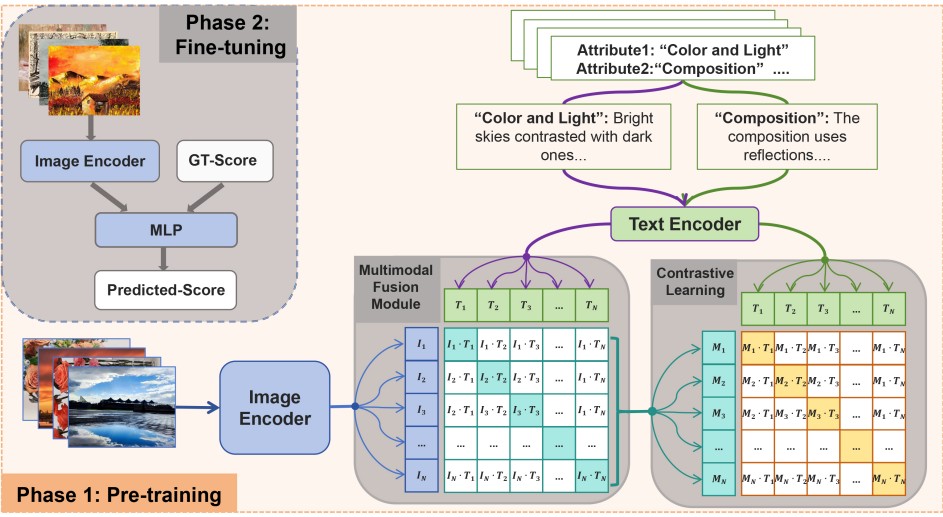

Figure 9: ArtCLIP samples two comments from different aesthetic attribute perspectives and further introduces a multimodal fusion module to integrate text and image embeddings, thereby generating multimodal embeddings for contrastive learning.

ArtCLIP is a pre-trained model designed to learn rich aesthetic concepts from a categorized multi-modal image aesthetic database. Its training process begins with attribute-aware learning, where diverse aesthetic attributes of images are selected as contrasting objects. Random sampling from corresponding attribute category comments is conducted to merge image embeddings with text embeddings, capturing semantics relevant to aesthetic attributes. Subsequently, contrastive learning is employed to bring positive pairs closer while pushing negative pairs apart, iteratively optimizing model parameters to acquire more general aesthetic concepts. Ultimately, the ArtCLIP model, equipped with learned aesthetic knowledge from pre-training, can be leveraged for various downstream tasks, offering abundant aesthetic information support for image processing and analysis.

The categorized DPC2022 dataset [Zhong et al., 2023] is the training set for ArtCLIP. Specifically, we have categorized the comments in DPC2022 into five aesthetic attributes: "depth and focus," "color and light," "subject of picture," "composition," and "use of camera," ensuring each image possesses at least two classification attributes. To apply the pre-trained ArtCLIP to painting aesthetic evaluation, fine-tuning is conducted on the APDDv2 dataset. Here, ArtCLIP's image encoder extracts feature vectors, followed by integration into an MLP network for score regression. The structure of the ArtCLIP network is illustrated in Figure 9.

During the pre-training phase of ArtCLIP, a batch size of 80 and 20 epochs were used, with an initial learning rate of 1e-4, and a linear learning rate decay adapter was applied. In the fine-tuning phase, the initial learning rate was set to 5e-5, the batch size to 64, and the total number of fine-tuning iterations to 10. All experiments were conducted on an NVIDIA GeForce RTX 4090 GPU.

For the comparative experiments, AANSPS[Jin et al., 2024] and SAAN[Yi et al., 2023] were trained on the APDDv2 dataset. The experimental results are presented in Table 4.

Table 4: Comparison of AANSPS, SAAN and ArtCLIP on APDDv2.

| Score Type | AANSPS MSE ↓ | MAE ↓ | SROCC ↑ | PLCC ↑ | ACC ↑ | SAAN MSE ↓ | MAE ↓ | SROCC ↑ | PLCC ↑ | ACC ↑ | ArtCLIP MSE ↓ | MAE ↓ | SROCC ↑ | PLCC ↑ | ACC ↑ |
|---|---|---|---|---|---|---|---|---|---|---|---|---|---|---|---|
| TAS | 0.88 | 0.73 | 0.76 | 0.79 | 0.89 | 1.79 | 0.99 | 0.78 | 0.61 | 0.86 | **0.68** | **0.63** | **0.81** | **0.84** | **0.89** |
| T&L | 0.73 | 0.68 | 0.70 | 0.72 | 0.87 | 1.98 | 1.07 | 0.48 | 0.49 | 0.83 | **0.60** | **0.60** | **0.74** | **0.77** | **0.87** |
| C | 0.81 | 0.72 | **0.71** | 0.72 | 0.85 | 1.84 | 1.05 | 0.48 | 0.49 | 0.78 | **0.71** | **0.67** | 0.74 | **0.74** | **0.85** |
| L&C | 0.74 | 0.68 | 0.74 | 0.77 | **0.89** | 1.49 | 0.93 | 0.56 | 0.58 | 0.82 | **0.63** | **0.61** | **0.77** | **0.80** | 0.88 |
| S&P | 0.76 | 0.70 | 0.72 | 0.79 | 0.91 | 1.60 | 0.95 | 0.60 | 0.63 | 0.85 | **0.60** | **0.61** | **0.79** | **0.83** | **0.91** |
| SO | 0.75 | 0.68 | 0.73 | 0.75 | 0.87 | 1.60 | 0.94 | 0.52 | 0.52 | 0.81 | **0.62** | **0.62** | **0.75** | **0.78** | **0.87** |
| L$S | 0.83 | 0.72 | 0.73 | 0.79 | 0.90 | 1.67 | 1.02 | 0.61 | 0.65 | 0.84 | **0.65** | **0.63** | **0.79** | **0.84** | **0.91** |
| Col | 0.80 | 0.70 | **0.79** | 0.78 | 0.91 | 1.76 | 1.00 | 0.54 | 0.60 | 0.89 | **0.59** | **0.59** | 0.75 | **0.84** | **0.92** |
| D&T | 0.90 | 0.74 | 0.76 | 0.78 | 0.86 | 1.62 | 0.97 | 0.62 | 0.62 | 0.82 | **0.70** | **0.65** | **0.81** | **0.83** | **0.88** |
| O | 0.79 | 0.70 | 0.73 | 0.77 | 0.89 | 1.35 | 0.89 | 0.58 | 0.62 | 0.85 | **0.63** | **0.62** | **0.78** | **0.81** | **0.89** |
| M | 0.88 | 0.74 | 0.71 | 0.73 | 0.86 | 1.83 | 1.02 | 0.52 | 0.53 | 0.80 | **0.71** | **0.67** | **0.75** | **0.78** | **0.85** |

| Score type | Image | Predicted | GT | Image | Predicted | GT | Image | Predicted | GT | Image | Predicted | GT |
|---|---|---|---|---|---|---|---|---|---|---|---|---|
| Total Score | | 84.5 | 82.5 | | 70.6 | 67.5 | | 51.2 | 51.3 | | 25.9 | 25 |
| T&L | | / | / | | 6.68 | 6.83 | | 5.43 | 5.12 | | 3.49 | 3.67 |
| Cre | | / | / | | 6.81 | 6.67 | | 5.18 | 4.75 | | / | / |
| L&C | | 8.04 | 8.00 | | 6.77 | 6.67 | | 5.52 | 4.88 | | 3.09 | 3.17 |
| S&P | | 7.93 | 8.00 | | / | / | | / | / | | 2.90 | 2.67 |
| SO | | / | / | | 6.58 | 6.83 | | 4.96 | 4.75 | | 3.02 | 3.00 |
| L&S | | 8.40 | 8.33 | | / | / | | / | / | | 2.65 | 2.67 |
| Col | | / | / | | 6.79 | 6.67 | | 4.86 | 4.38 | | 2.99 | 3.00 |
| D&T | | 8.65 | 7.67 | | 6.88 | 6.83 | | 4.66 | 4.5 | | 2.43 | 2.33 |
| TO | | 8.25 | 8.17 | | 7.01 | 7.00 | | 4.72 | 4.88 | | 3.14 | 3.33 |
| M | | / | / | | / | / | | 4.92 | 5.12 | | 2.65 | 2.67 |

Figure 10: Test samples. *Predicted* represents the predicted score of the ArtCLIP output. *GT* represents the ground-truth score.

# 5 Prospects for APDDv2 and ArtCLIP applications

The integration of APDDv2 and ArtCLIP provides a powerful multimodal platform for the aesthetic assessment of artistic images. As the first multimodal dataset featuring detailed aesthetic language commentary, APDDv2 encompasses a diverse array of artistic styles and aesthetic attributes, making it highly applicable for multimodal learning models. By combining visual and textual information, the model enhances both the understanding and generation of aesthetic features. Moreover, APDDv2

can also be utilized for sentiment analysis, allowing for the evaluation of emotional responses elicited by artworks through the analysis of comments and ratings.

ArtCLIP, as a multimodal scoring network architecture, has been trained to develop scoring models based on the APDDv2 dataset. Our team is collaborating with several art institutions to investigate the feasibility of applying this scoring model in teacher-assisted instruction and children's art education. Additionally, the ArtCLIP model is capable of regulating the aesthetic quality of images generated by AI-generated content (AIGC), thus supporting tasks such as image generation and style transfer.

# 6 Limitations

We acknowledge that while APDDv2 offers a diverse and comprehensive set of annotations, it has limitations in fully encompassing all artistic styles and aesthetic preferences, particularly in underrepresented or emerging art genres. Furthermore, although the evaluation criteria are robust, they may require further refinement to account for subjective variations in aesthetic judgment across different cultural contexts. Our dataset was intentionally designed to focus on expert-level aesthetic evaluations in order to ensure the professionalism and depth of the scores and comments. Consequently, we did not initially consider incorporating public aesthetic opinions. However, we recognize the importance of broader perspectives and therefore plan to include public evaluations in future expansions of the dataset. While we carefully considered factors such as age, position, painting experience, and educational background when selecting annotators to mitigate cultural biases, we recognize that this is still insufficient. In the future, we will include annotators from diverse countries and cultural backgrounds to further reduce cultural biases and enhance the dataset's applicability.

# 7 Conclusions

Over the course of more than a year, we have assembled a diverse group of over 70 experts, teachers, and students from the fields of painting and art to engage in the extensive task of data collection and fine-grained annotation. Ultimately, we have successfully constructed the first dataset in the painting domain that exceeds 10,000 images, covering 24 artistic categories and exploring 10 aesthetic attributes in depth, while also integrating abundant linguistic commentary information.

On this foundation, we further trained the ArtCLIP model on the APDDv2 dataset. This model is capable of comprehensively evaluating the total aesthetic value of artistic images and scoring various aesthetic attributes. Through experimental validation, the ArtCLIP model has demonstrated superior performance compared to existing artistic image scoring models.

Looking forward, we plan to further expand the scale and diversity of the APDDv2 dataset, increasing the variety of images and incorporating more linguistic commentary during the labeling process. We even aim to introduce more elaborate annotation forms such as image annotation circles. It is worth noting that due to the scarcity of painting commentary datasets, we had to utilize photography commentary datasets as an alternative during the pre-training of the ArtCLIP model. However, as our research progresses, we will strive to construct a dedicated artistic image commentary dataset and incorporate a larger-scale painting scoring dataset to refine the training of the ArtCLIP model, aiming to achieve even more outstanding performance in the field of artistic aesthetic evaluation.

## Acknowledgments and Disclosure of Funding

We thank the ACs and reviewers. This work is partially supported by the Natural Science Foundation of China (62072014), the Fundamental Research Funds for the Central Universities (3282024049). We would also like to thank the annotators, including Guangdong Li, Jie Wang, Tingting Ma, Xingming Liu, and others from institutions such as the Central Academy of Fine Arts, Taiyuan Normal University, Beijing Institute of Fashion Technology, etc.

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
