# OpenReview forum: "APDDv2:  Aesthetics of Paintings and Drawings Dataset with Artist Labeled Scores and Comments"
_NeurIPS.cc/2024/Datasets_and_Benchmarks_Track — NeurIPS 2024 Track Datasets and Benchmarks Poster_

### Official Review · Reviewer_pGpG · 2024-07-22
**Comprehensive Art Dataset**

**Rating:** 6
**Confidence:** 3
**Clarity:** The paper is well written.

**Review:**

The interesting part of the proposed dataset is the engagement with Domain experts. Adding their knowledge into the annotation makes the dataset APDDv2, becoming fairly more interesting than APDDv1. The dataset acquisition and categorisation is well described and compared with previous datasets in the same fields have more content. Distinguish between these Aesthetic Attributes is the most elaborated part of the dataset, decomposing the dataset in terms of complexity. This element could be used for further experiments on LLM reasoning or symbolic machine learning. Perhaps, the authors didn't focus on this aspects. They focused on the Machine learning categorisation of their dataset. Indeed, they show that the fine-tuned ArtCLIP can distinguish particularly well the different categories, making the dataset not challenging for machine learning benchmarks. The dataset is interesting but the authors did not focus on the application.

**Strengths:**

- Dataset with domain experts annotations
- Comprehensive art categorisation

**Additional Feedback:**

No additional comments.

**Correctness:**

The dataset is well constructed and the claims agree with the results. Some evaluations are missing.

**Documentation:**

The dataset is well described and available publicly.

**Ethics:**

No concerns about the ethics.

**Limitations:**

- Lack of evaluation metrics. They compared with the model introduced by their previous paper APDDv1

**Opportunities For Improvement:**

- Focus on the application of the dataset.
- Benchmark the dataset with more pre-trained Vision Language models.

**Relation To Prior Work:**

The dataset is an extension of an existing dataset. The dataset has been extended from 5000 to 10000 images with further labels descriptions. Related works are well-covered.

**Summary And Contributions:**

The authors introduce the extension of an existing art datasets with categories, attributes and description. The dataset focuses on Oil paintings, Sketches and Traditional Chinese paintings. The dataset is benchmarked with the fine-tuning of a pre-trained ArtCLIP (a variation of CLIP for art datasets).

---

> ### Author Rebuttal · Authors · 2024-08-10
>
> We greatly appreciate your positive feedback and for pointing out areas where the paper could be improved！
>
> As stated in the paper, the primary application of our dataset is in training aesthetic scoring model  ArtCLIP. We are actively applying the model in teacher-assisted instruction and children's art education. We will provide a clearer explanation of the dataset's applications in the final version of the paper and discuss other potential applications, such as image generation, style transfer, and more.
>
> Additionally, in the final version, we will benchmark the dataset using more pre-trained Vision Language models, such as DALL-E, BLIP, and others, to further validate both the dataset's usability and the effectiveness of the models.
>
> Regarding the experimental comparisons, in addition to the model introduced in APDDv1, we also compared our model with the SAAN model published at CVPR 2023. The results demonstrate the effectiveness of the ArtCLIP model.
>
> Thank you once again for your valuable time and constructive suggestions.

---

### Official Review · Reviewer_WijH · 2024-07-23
**Meaningful painting dataset for aesthetics**

**Rating:** 7
**Confidence:** 4
**Correctness:** correct
**Clarity:** The paper written is clear

**Review:**

The dataset's expansion and enriched annotations demonstrate a meticulous approach to improving the precision and depth of aesthetic evaluation, addressing a notable gap in the domain of artistic image analysis.

The clear articulation of methodologies and the structured presentation of results enhance the comprehensibility of the paper, making it accessible to a broad audience.

Furthermore, the development of the ArtCLIP model, underscores the paper's substantial contribution to advancing the field of aesthetic evaluations in art, providing valuable tools and resources for both academic research and practical applications.

**Strengths:**

The APDDv2 dataset offers a broad and diverse range of images. This comprehensive coverage allows for more nuanced and detailed analysis of various art forms and styles.

With over 90,000 annotations and detailed language comments provided by experts, the dataset ensures high-quality, precise, and contextually rich evaluations.

**Additional Feedback:**

N/A

**Documentation:**

Github related to the paper is well maintained.

**Ethics:**

no ethical concerns.

**Limitations:**

no potential negative societal impact

**Opportunities For Improvement:**

Even with expert annotations, aesthetic evaluations are inherently subjective. Different annotators might have varying opinions on what constitutes aesthetic quality, which could introduce inconsistencies and biases in the dataset. For example, how do the general public people evaluate the aesthetics other than experts?

The dataset might reflect the annotators' cultural biases and the artworks' selection. Art and aesthetics are deeply influenced by cultural contexts, and a dataset predominantly featuring one cultural perspective might not be universally applicable.

**Relation To Prior Work:**

Discussed

**Summary And Contributions:**

The paper introduces an enhanced dataset for evaluating the aesthetics of artistic images. The APDDv2 dataset expands upon its predecessor, APDDv1, by increasing the image count and improving annotation quality with detailed language comments.

It encompasses many artistic categories and multi-dimensional aesthetic attributes, providing a comprehensive resource for researchers. Additionally, the paper presents ArtCLIP, a refined aesthetic evaluation model that surpasses previous models in accuracy and effectiveness.

The dataset and model are accessible at APDDv2 GitHub.

---

> ### Author Rebuttal · Authors · 2024-08-10
>
> Thank you very much for your valuable review and feedback. We greatly appreciate your recognition of our work! We also sincerely thank you for your insightful feedback on the subjectivity of aesthetic evaluations and the potential cultural biases in our dataset.
>
> Our dataset was intentionally designed to focus on expert-level aesthetic evaluations, with the primary goal of ensuring the professionalism and depth of the scores and comments. As such, we did not initially consider incorporating public aesthetic opinions. However, we acknowledge the importance of broader perspectives, and we plan to include public aesthetic evaluations in future expansions of the dataset.
>
> To minimize the impact of cultural biases on the dataset, we have carefully considered factors such as age, position, painting experience, and educational background when selecting annotators. Moving forward, we aim to further enhance the dataset's applicability by incorporating annotators from diverse countries and cultural backgrounds.
>
> We appreciate your comments, and we believe these steps will help address the concerns raised and contribute to the continuous improvement of our work.

---

### Official Review · Reviewer_bTSD · 2024-07-26
**Valuable contribution**

**Rating:** 7
**Confidence:** 4
**Correctness:** Yes.
**Clarity:** The paper is clearly motivated and wr…

**Review:**

# Strengths

Overall, this is a great paper that is:

- Clearly motivated and written.
- Has a meticulous approach to data collection and annotation, including the use of expert raters.
- Evaluated in terms of qualitative distribution of ratings as well as through training of a CLIP style model.
- Has the potential for significant impact in driving research in the evaluation, and thus ultimately in the synthetic generation, of art.

# Weaknesses

The only missing analysis I found in the paper was of variance in scores between different raters. The paper acknowledges that assessing creative works is inherently subjective. The dataset assigns "at least" 6 raters per example. But, it would be good to understand if 6 raters are enough --- what is the variance in ratings, and how does it differ by category and attribute?

It would also be interesting to evaluate the variance in free-form comments. This is of course more tricky --- but perhaps a cross-validation strategy with training a CLIP model could be employed---for example, train CLIP models leaving out one rater, and then see if that model can accurately match the left out raters' comments to the right example.

Note that the evaluation of comment-variance would be "nice to have", but I believe variance of numeric ratings should be included in the paper.

**Strengths:**

See above.

**Additional Feedback:**

See above.

**Documentation:**

Description is clear.

**Ethics:**

No ethical concerns.

**Limitations:**

See above.

**Opportunities For Improvement:**

See above.

**Relation To Prior Work:**

References are adequate.

**Summary And Contributions:**

The paper introduces a new dataset of paintings with expert annotations to accurately evaluate models that assess artwork. The dataset contains various categories of paintings, with per-category attribute scores as well as free-form comments to capture image quality.

---

> ### Author Rebuttal · Authors · 2024-08-10
>
> Thank you very much for your positive feedback on our paper and for your valuable suggestions! We particularly appreciate your reminder to consider the variance in scores between different raters, which is indeed an important analytical angle.
>
> In response to your suggestion, we have calculated the variance in total aesthetic scores for 24 categories of images. With the scores normalized to a maximum of 10, we found that for these 24 categories, 17 categories have a variance less than 3, and the overall average variance across categories is 2.72. Based on the AVA dataset, we consider that a variance below 3 typically indicates high consistency and reliability in the ratings. However, we also observed that the highest average variance exceeds 3.5, indicating that there may be some outlier data.
>
> To improve the usability and reliability of our dataset, we plan to exclude these data with higher variance in the final version. Additionally, we will include a detailed discussion on the variance of the aesthetic scores for the 10 attributes across the 24 image categories in the paper to further clarify the reliability of the data and its impact on aesthetic evaluation.
>
> Evaluating the variance in free-form comments and employing a cross-validation strategy with CLIP models is indeed an intriguing and practical direction. This approach could enhance the depth of analysis of the comment data and the reliability of the model. We appreciate your suggestion and will explore this method in future dataset and model research.
>
> Once again, thank you for taking the time to provide such valuable feedback on our paper!

---

> > ### Comment · Reviewer_bTSD · 2024-08-30
> > **Thanks!**
> >
> > Thanks for the explanations. I believe the promised discussion on variance will definitely improve the paper's impact. Happy to keep my positive score after reading the other reviews as well.

---

### Official Review · Reviewer_UxjQ · 2024-08-03
**Aesthetic Painting and Drawing Dataset v2**

**Rating:** 7
**Confidence:** 3

**Review:**

The paper improves on the APDD dataset by adding more images and annotations to the dataset. Additionally a CLIP model is trained on the new dataset to assess the aesthetic quality of drawings and paintings. This is a relevant dataset with good documentation and enough samples. However, I personally feel that that the problem of judging the aesthetic quality of images has real-world impact.

**Strengths:**

- The paper is well written and presented.
- The dataset has sufficient number of images and annotations
- The data collection and annotation process is clearly documented.

**Additional Feedback:**

- See Opportunities for Improvement section.

**Clarity:**

- The paper is well written and presented.

**Correctness:**

- The content of the paper is correct and the data collection and annotation steps look correct as well.

**Documentation:**

- The data creation and annottion process in well documented. The dataset is available publicly and has sufficient documentation.

**Ethics:**

- There are no significant ethical concerns with the paper.

**Limitations:**

- The paper doen't discuss the limitations of the proposed dataset and approach.

**Opportunities For Improvement:**

- I would suggest the authors to try and focus more on problems with greater real world impact. but, overall great efforts on this paper.

**Relation To Prior Work:**

- the prior aesthetic datasets and the changes in this new dataset as compared to the others are clearly presented.

**Summary And Contributions:**

The paper introduces a new verision of the Aesthetics Painting and Drawing Dataset, which they call APDDv2. The dataset consists of 10,000 images with almost 90,000 annotations. The goal of the dataset is to evaluate aesthetic quality of paintings and drawings. The eisting aesthtic datasets focus mainly on general photos and not on paintings and drawings. Additionally, the authors train an ArtCLIP model on the new dataset. Each image is labeled on 10 aethetic attributes.

---

> ### Author Rebuttal · Authors · 2024-08-10
>
> Thank you very much for your valuable feedback and insights! We sincerely appreciate your recognition of our work and will carefully consider your suggestions to further improve our paper and future research.
>
> We acknowledge that while APDDv2 provides a diverse and comprehensive set of annotations, it may still have limitations in fully encompassing all artistic styles and aesthetic preferences, particularly in underrepresented or emerging art genres. Additionally, while the evaluation criteria are robust, they may require further refinement to account for subjective variations in aesthetic judgment across different cultural contexts. Extending the ArtCLIP model to styles or artworks that significantly deviate from those represented in the dataset will also pose challenges. In the final version of our paper, we will include a dedicated section discussing the limitations of both the dataset and the approach.
>
> Once again, thank you for taking the time to provide such valuable feedback!

---

### Decision · Program_Chairs · 2024-09-26

**Decision:**

Accept (Poster)

**Comment:**

The paper introduces APDDv2, an expanded dataset of 10,000 paintings and drawings with approximately 90,000 expert annotations evaluating aesthetic quality across 10 attributes. It also presents ArtCLIP, a model trained on this dataset for aesthetic evaluation.

Strengths from the reviewers
+ Well-documented dataset with expert annotations: the dataset is meticulously constructed, enhancing depth and quality. (R-UxjQ, R-bTSD, R-WijH, R-pGpG)
+ Comprehensive artistic coverage: APDDv2 offers nuanced analysis across various art forms. (R-WijH, R-pGpG)
+ Clear presentation: the paper is well-written and accessible. (R-UxjQ, R-bTSD, R-WijH, R-pGpG)
+ Potential impact on art evaluation research: the work could advance aesthetic evaluation and art generation. (R-bTSD)

Weaknesses:
- Subjectivity and cultural bias: annotations may reflect biases; inclusion of public opinions is suggested. (R-WijH)
- Missing rater variance analysis: lack of analysis on score variance raises reliability questions. (R-bTSD)
- Limited applications and benchmarks: more focus on practical applications and additional model benchmarking is needed. (R-pGpG)
- No limitations discussion: the paper doesn't address its own limitations. (R-UxjQ)

In the rebuttal the authors plan to discuss limitations, address subjectivity and biases, analyze rater variance, and enhance benchmarking and applications in future work. All the reviewers vote for accept and hence the AC recommends acceptance.